# Machine Learning meets Algebraic Combinatorics: A Suite of Datasets to Accelerate AI for Mathematics Research

**Herman Chau**[†*]**, Helen Jenne**[‡,*]**, Davis Brown**[‡]**, Jesse He**[¶,‡]**,**
**Mark Raugas**[‡]**, Sara Billey**[†]**, Henry Kvinge**[‡,†]

[†] University of Washington
[‡] Pacific Northwest National Laboratory
[¶] University of California, San Diego
[*] equal contribution
hchau@uw.edu, henry.kvinge@pnnl.gov

## Abstract

The use of benchmark datasets has become an important engine of progress in machine learning (ML) over the past 15 years. Recently there has been growing interest in utilizing machine learning to drive advances in research-level mathematics. However, off-the-shelf solutions often fail to deliver the types of insights required by mathematicians. This suggests the need for new ML methods specifically designed with mathematics in mind. The question then is: what datasets should the community use to drive this research? On the one hand, toy problems such as learning the multiplicative structure of small finite groups have become popular in the mechanistic interpretability community whose perspective on explainability aligns well with the needs of mathematicians. While toy datasets are a useful to guide initial work, they lack the scale, complexity, and sophistication of many of the principal objects of study in modern mathematics. To address this, we introduce a new collection of datasets, the Algebraic Combinatorics Dataset Repository (ACD Repo), representing either classic or open problems in algebraic combinatorics, a subfield of mathematics that studies discrete structures arising from abstract algebra. After describing the datasets, we discuss the challenges involved in constructing"good" mathematics dataset for ML and describe baseline model performance.

## 1  Introduction

Modern approaches to machine learning (ML) have been shown to be capable of extracting sophisticated patterns from large and complex datasets. As this capability has grown, there is increasing evidence that ML can be used as a tool to advance scientific discovery. Beyond the natural sciences, a growing community of researchers are also looking at ML as a tool to aid the research mathematician. Some of this research explores the use of LLMs and related models to aid in proof writing and higher mathematical reasoning [26, 33], but there is also a need for models to help analyze (what we call) 'raw' mathematical data. This data, which often takes the form of (very) long lists of examples, is used by the mathematician to develop intuition and formulate conjectures. Though the popular perception is that research mathematics takes place at a level of abstraction beyond individual instances, the manual examination of examples (data) constitutes a fundamental part of the mathematician's workflow for many problems. For example, when trying to better understand the coefficients of a particular family of polynomials (e.g., Kazhdan-Lusztig polynomials in Section 2), a mathematician

may look through countless examples to either generate conjectures or build evidence for existing conjectures that they have.

Existing applications of machine learning to raw mathematics data tend to fall into one of two types. The first are toy problems that are simple enough that we can hope to explicitly describe how the model is solving the problem [35, 24, 23, 22]. These are used by AI researchers in the interpretability community as small, self-contained, and tractable problems where at least one solution is very well understood. By design these do not represent (or even aim at) open problems in mathematics. The other cluster of works focus on solving very specific problems in research-level mathematics [11, 32, 18, 6, 18, 12]. These works tend to come from the mathematics community and often have a high knowledge barrier, requiring the reader to already be familiar with the underlying domain.

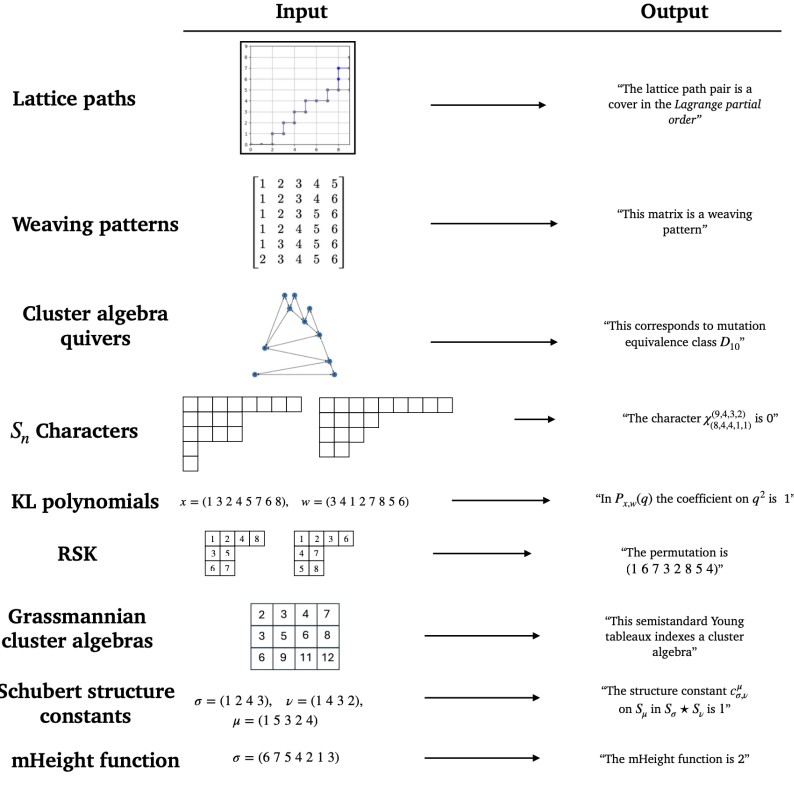

Figure 1: A visualization of some of the tasks included in these datasets.

We noticed that there is a need for research-level mathematics datasets accessible to the broader ML community. To fill the gap we present the *Algebraic Combinatorics Dataset Repository (ACD Repo)*[1], a collection of 9 datasets designed for the development of machine learning tools to advance research mathematics. Our collection includes both open problems and classic problems whose solution is a major result in the field. Algebraic combinatorics is an area of mathematics that studies discrete structures arising from abstract algebra (including algebraic geometry and representation theory). We chose to focus on this domain because: (i) it requires less background theory to understand, making it generally more accessible than, for instance, algebraic topology, differential geometry, or analysis, while still remaining absolutely fundamental to cutting-edge mathematics, (ii) there already exist specialized software libraries (e.g., Sage [29]) designed to efficiently compute many quantities of interest in algebraic combinatorics, and (iii) by nature of being discrete, the objects of interest in algebraic combinatorics tend to be more amendable to representation on a computer.

We believe that three communities will find this collection useful. The most obvious are researchers from either machine learning or mathematics that are interested in accessible problems that can be used when developing methods for mathematics research. The second is interpretability researchers,

---

[1]Datasets and associated code can be found at `https://github.com/pnnl/ML4AlgComb`

since finding a solution to an open problem through the interpretation of a performant neural network would be a major accomplishment for the field. Finally,'science of deep learning' researchers may be interested in datasets representing complex tasks for which one can generate an arbitrary amount of data and make data of arbitrary input dimension and complexity, adding to existing resources in this direction [30].

## 2   Dataset Descriptions

In this section we provide very abbreviated descriptions of the datasets currently available in the repository. Further details including: problem context, how the datasets were generated, and a datasheet for each dataset can be found in Section C. For a summary statistics table, see Table D.1. Background on algebraic combinatorics and its associated data structures can be found in Section B.

1. **Partial orders on lattice paths (open problem):** Pairs of NE lattice paths $(w, w')$ on a grid of size $n \times n - 1$ where $w'$ covers $w$ in either the matching or Lagrange order (but not both). We include $n = 10, 11, 12$. **Task:** Train a model that can predict whether $(w, w')$ is a covering pair in Lagrange or matching order. The ultimate goal is to better understand Lagrange and matching order.

2. **Weaving patterns (open problem):** A mixture of enriched weaving patterns and non-weaving pattern matrices with $\{1, 2, \ldots, n\}$-entries. We include matrices of size $6 \times 5$, $7 \times 6$, and $8 \times 7$. **Task:** Classify whether a matrix in the dataset is a weaving pattern or not. The ultimate goal is to come up with a succinct characterization of weaving patterns.

3. **Mutation equivalence of quivers (open problem):** Adjacency matrices for quivers, each with 11 vertices, labeled by mutation equivalence class. **Task:** Classify which mutation equivalence class an adjacency matrix corresponds to. The goal is to find simple characterizations of mutation equivalence classes where they are not already known.

4. **Symmetric group characters (classic result):** Pairs of integer partitions of $n$, $\lambda, \mu$ and the corresponding symmetric group character $\chi_\mu^\lambda$. We include $n = 18, 20, 22$. **Task:** Given partitions $\lambda$ and $\mu$, predict the irreducible symmetric group character $\chi_\mu^\lambda$. Several methods of computing this are already known. New approaches would be of high value to the community.

5. **The coefficients of Kazhdan-Lusztig polynomials (open problem):** Each instance in this dataset consists of a pair of permutations on $n$, $\sigma$ and $\nu$, along with the coefficients of the Kazhdan-Lusztig polynomial $P_{\sigma,\nu}(q)$. We provide $n = 8, 9$. **Task:** Predict the coefficients of $P_{\sigma,\nu}(q)$ given $\sigma$ and $\nu$. These coefficients on $P_{\sigma,\nu}(q)$ are known to carry algebraic and geometric information.

6. **The mHeight function of a permutation (intermediary result):** Permutations of size $n$ labeled by their mHeight. We provide datasets for $n = 9, 10$. **Task:** Predict the mHeight of a permutation. This was an important intermediate step in the recent solution [14] of an open conjecture [7].

7. **Schubert polynomial structure constants (open problem):** Each instance in this dataset is a triple of permutations $(\beta, \gamma, \alpha)$, labeled by its structure constant $c_{\beta\gamma}^\alpha$ in the expansion of the product of Schubert polynomials $\mathfrak{S}_\beta \mathfrak{S}_\gamma$. Not all possible triples of permutations are included; the dataset consists of an approximately equal number of zero and nonzero coefficients. We provide datasets for $n = 4, 5, 6$. **Task:** Predict the coefficient $c_{\beta\gamma}^\alpha$. Better understanding Schubert polynomial structure constants is a longstanding open problem.

8. **The Robinson-Schensted-Knuth correspondence (classic result):** This dataset consists of triples: two standard Young tableau of size $n$ and their corresponding permutation (via the RSK algorithm). We include datasets for $n = 8, 9, 10$. **Task:** Given pairs of standard Young tableau, predict the corresponding permutation. This is a fundamental tool in algebraic combinatorics and it would be interesting to see if it could be rediscovered via ML.

9. **Grassmannian Cluster Algebras and Semistandard Young Tableau (open problem):** A collection of rectangular semistandard Young tableau each with a label indicating whether they index a cluster variable or not. **Task:** Predict whether a Young tableaux indexes a cluster variable.

# 3 Discussion

**Challenges when generating mathematics datasets:** Generating useful datasets for mathematics problems presents a number of challenges. For instance, imbalance is an issue in several different respects. On the one hand, traditional class imbalance comes up frequently, but there can also be imbalance in terms of how interesting the examples of a dataset are. For a given task, it may be the case that the vast majority of randomly sampled instances are uninteresting because they can be predicted or classified for straightforward reasons. In these cases, individual instances do not capture the mathematics that we care about. When enough data exists, one way to mitigate this situation is to subsample for harder examples. This is what we did for a number of the datasets in the ACD Repo including Weaving Patterns where we imposed some additional constraints on the non-weaving pattern $\{1, 2, \ldots, n\}$-matrices to make them harder to distinguish from true weaving patterns. The choice of input representation can also have large downstream impacts on how hard it is for a model to learn to solve a task. For example, there are many equivalent ways to represent a permutation. Similar to other parity prediction tasks [16], prediction of permutation parity is a hard task for transformers when the permutation is presented in one-line notation. Models do substantially better when input permutations are represented via their inversion vector.

**Baselines:** We ran our initial baselines on all datasets in the repository using logistic regression, MLPs, and encoder-only transformers. Our hyperparameter optimization strategy is detailed in Section D.1 and our results are summarized in Table 1. Our goal was to understand how well standard architectures and training techniques worked for these tasks. As such, we did not use specialized architectures or tune the data representations. We expect that doing this would substantially increase performance across the board. As can be seen, the performance of models varies considerably on different tasks. Some tasks are quite hard (such as symmetric group character calculation) with results that are nearly equivalent to guessing the most populous class. Other tasks were considerably easier such as predicting the mHeight function of a permutation. In general, MLPs tended to perform best on average without specific tuning, though transformers tend to catch up as the datasets become larger. We provide a longer discussion of model performance in Section D.

**Dependence on $n$:** Many problems in algebraic combinatorics have a natural dependence on a parameter $n$ (e.g., permutations are parametrized by the number of elements that they permute). We have chosen to structure datasets in the ACD Repo to reflect this, with the majority of datasets taking the form of a series of datasets $\{D_n\}_{n \geq 1}$. We provide a few values of $n$ and, in many cases, the code to generate others. Generally, there are two properties that change as $n \to \infty$. First, the size of $D_n$ grows as $n$ grows. The rate of growth depends on the specific problem, with many $|D_n|$ growing exponentially (such as those datasets that depend on the number of permutations of $n$). On the other hand, the problems also tend to become more complex.

Experimentally we have found that larger values of $n$ tend to lead to better model performance. For example, we ran 5 2-layer MLP models for 500 epochs on the *Lattice Path Datasets* corresponding to grids of size $6 \times 5, \ldots, 13 \times 12$. We see in Figure 8 (left) that with an interesting exception of moving from $7 \times 6$ to $8 \times 7$, performance across a range of dimensions improves as $n$ grows. There are exceptions however. We also looked at sampling from greater depth when exploring the Mutation Equivalent Quiver dataset (this means allowing a greater number of mutations to be applied to the initial quiver). As shown in Figure 8 (center), we find that performance somewhat degrades even though the size of the datasets increases. We suspect that exploration of the complexity of these problems (where it is known) might be an avenue for shedding light on this phenomenon.

# 4 Conclusion and Limitations

In this paper we introduced Algebraic Combinatorics Dataset Repository, a collection of datasets structured for machine learning and designed to facilitate the development of machine learning methods for advancing research level mathematics. While we believe that these datasets will provide significant value to the ML community, they also have some limitations. While we think we made reasonable choices, the novelty of the field of AI for math means that we can't be certain of this. Despite these limitations, we believe that ML tools for mathematics is a promising route to a richer and more diverse mathematics. We hope that these these will be useful to researchers looking to make progress in this area.

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

| Dataset | Logistic regression | MLP | Transformer |
|---|---|---|---|
| Lattice paths | | | |
| $n = 10$ | 66.2% | 84.2 % ± 1.3% | 71.0 % ± 1.6% |
| $n = 11$ | 66.3% | 90.5 % ± 1.6% | 66.7 % ± 0.3% |
| $n = 12$ | 66.5% | 96.2 % ± 1.0% | 86.8% ± 2.0% |
| Weaving patterns | | | |
| $n = 6$ | 70.4% | 84.5 % ± 1.0% | 83.4% ± 1.0% |
| $n = 7$ | 85.8% | 98.7 % ± 0.8% | 98.1 % ± 1.0% |
| Cluster algebra quivers | 40.3 % | 80.0 % ± 1.8% | 79.0%± 2.5% |
| Grassmanian cluster algebras | 65.7% | 98.8 % ± 0.4% | 98.4 %± 1.0% |
| KL polynomials | | | |
| $n = 8$ | 78.2% | 84.1 % ± 1.1% | 83.0% ± 1.0% |
| $n = 9$ | 57.0% | 74.9 % ± 1.5% | 92.4% ± 1.6% |
| Schubert polynomials | | | |
| $n = 4$ | 64.4% | 86.9 % ± 1.7% | 86.1% ± 3.8% |
| $n = 5$ | 66.7% | 94.8 % ± 2.6% | 78.0% ± 2.5% |
| $n = 6$ | 65.5% | 97.3% ± 0.00% | 81.1% ± 2.6% |
| mHeight | | | |
| $n = 9$ | 70.8% | 89.6 % ± 1.8% | 79.3% ± 2.2% |
| $n = 10$ | 94.2% | 99.4 % ± 0.4% | 98.7% ± 0.5% |

Table 1: Naive baseline model accuracy on classification datasets. Results are averaged over three random weight initializations with 95% confidence intervals.

| Dataset | Linear regression | MLP | Transformer |
|---|---|---|---|
| $S_n$ characters | | | |
| $n = 18$ | 1,101.9 | 1,118.2 ± 88.2 | 7,487.8 ± 4,834.2 |
| $n = 20$ | 3,915.9 | 3,917.7 ± 54.4 | 27,885.2 ± 18,542.1 |
| $n = 22$ | 13,608.9 | 30,504.5 ± 31,690.4 | 73,957.0 ± 60,522.2 |
| RSK | | | |
| $n = 8$ | 0.2053 | 1.5 ± 0.4 | 2.7 ± 0.3 |
| $n = 9$ | 0.2133 | 2.3 ± 0.3 | 3.4 ± 0.5 |

Table 2: Naive baseline model mean squared error (MSE) on regression datasets. Results are averaged over three random weight initializations with 95% confidence intervals.

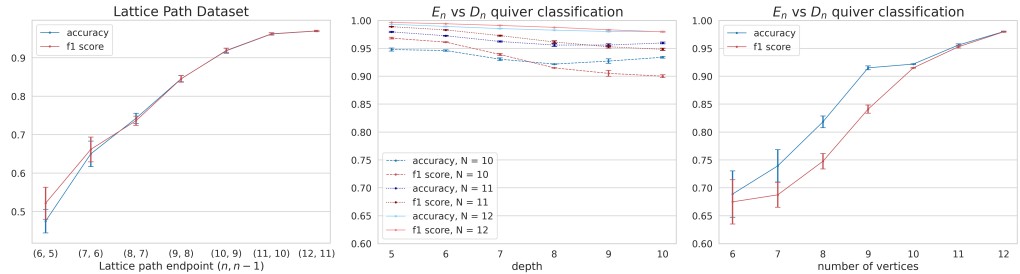

Figure 2: **(Left)** Performance on the *Lattice Path Dataset* as a function of the width of the $n \times n - 1$ grid on which lattice paths are constrained to. As $n$ grows in $n \times n - 1$, the training set size increases but problem complexity may also grow. **(Center)** Performance on the type $E$ versus type $D$ quiver classification task as a function of the depth, which must be specified for type $E$ quivers on $N = 10, 11, 12$ vertices, and **(Right)** the number of vertices $N$.

## A   Related Work

**AI for Mathematics:** There has recently been a growing number of papers that use machine learning based methods to assist in mathematics research. While many of these focus more on the proof-creation part of the mathematician's workflow [26, 33, 4], there are also many that look at the raw mathematical data. This includes the search for counterexamples in graph theory [32], the search

for connections between different knot invariants [12], the classification of $\mathbb{Q}$-Fano varieties [11], and Clifford invariants of ADE Coxeter elements [9]. Unlike these works which aim to shed light on specific problems, our work's goal is to generate datasets so that both the expert and non-expert can apply machine learning to a range of research level mathematics problems.

**Neural Algorithmic Reasoning:** This new field of machine learning looks at applying machine learning methods to algorithmic data [31]. Like mathematics, applying machine learning to algorithmic data provides a setting where arbitrarily large amounts of data can be generated. Further, as with mathematics, working with algorithmic data allows us to manipulate certain aspects of the problem (e.g., complexity) in ways not possible in noisy real-world data. The differences between this work and the primary benchmark for ML applied to classical algorithmic tasks, [30], is substantial. Most notably, the focus of these datasets is not on scientific discovery like the ACD Repo. Finally, the types of problem and many of the datatypes in the present work have not been investigated within the neural algorithmic reasoning community.

Permutation
$$\sigma = 3\ 4\ 5\ 1\ 2$$

**3 4 1 2** patterns
$$3\ 4\ 1\ 2 \quad 4\ 5\ 1\ 2 \quad 3\ 5\ 1\ 2$$

mHeight of $\sigma$
$$\max(3-2, 4-2, 3-1) = 2$$

Figure 3: An example of the calculation of mHeight on a permutation.

# B    Background and Datatypes

The field of combinatorics studies a broad range of problems in mathematics centered around discrete objects (e.g, partial orders, graphs, permutations, partitions) [27, 28]. Ideas and tools from combinatorics play an essential role in many other fields of mathematics and continue to have a strong impact on computer science and physics. Algebraic combinatorics is a subfield of combinatorics that applies combinatorial methods to problems arising from abstract algebra, particularly algebraic geometry and representation theory. In this section we review datatypes that play a central role in the field and that appear in the ACD Repo.

**Partitions:** We use the word partition in this work to mean an integer partition. An integer partition of $n \in \mathbb{N}$ is a sequence of positive integers $(n_1, n_2, \ldots, n_k)$ such that $n = n_1 + n_2 + \cdots + n_k$ and $n_1 \geq n_2 \geq \cdots \geq n_k$. We use the standard notation $\mu \vdash n$ to denote that $\mu$ *is a partition of* $n$. A partition $(n_1, \ldots, n_k)$ is often visualized as a *Young diagram*, with $n_1$ left justified square cells in the first row, $n_2$ left justified square cells in the second row, etc. See Figure 4 (left) for an example of a Young diagram corresponding to the partition $(3, 2, 2)$.

| | | |
|---|---|---|
| | | |
| | | |

| 7 | 6 | 4 |
|---|---|---|
| 5 | 3 | |
| 2 | 1 | |

| 1 | 2 | 4 |
|---|---|---|
| 3 | 5 | |
| 6 | 7 | |

Figure 4: **(Left)** A Young diagram for the partition $(3, 2, 2)$. **(Center)** A standard Young tableaux for the partition $(3, 2, 2)$. **(Right)** A semistandard Young tableaux for the partition $(3, 2, 2)$.

**Young tableaux:** We noted above that a partition can be visualized as a Young diagram. Surprisingly, including extra decorations on the cells in a Young diagram can capture fundamental combinatorics in representation theory and other fields. A *Young tableau* corresponding to a Young diagram $\lambda \vdash n$ is a labeling of the cells of $\lambda$ by an alphabet of symbols. In this work we will consider two types of Young tableau. A *standard Young tableau* corresponding to partition $\lambda \vdash n$ is a labeling of the cells of $\lambda$ by $1, 2, \ldots, n$ such that the integers strictly increase as one moves down a column or left to right across a row. See Figure 4 (center) for an example of a standard Young tableau for the partition

$(3, 2, 2)$ of 7. The definition of a *semistandard Young tableau* is analogous except that the entries are only assumed to weakly increase as one moves from left to right along a row (see Figure 4 (right)).

**Permutations:** Permutations are familiar in machine learning from their central role in computer science as well as their relevance to symmetries in many neural networks [13, 2, 15] and as a symmetry in graphs- [20] and set-based problems [34, 21]. There are many ways to represent a permutation. In this paper we use *one-line notation*, which is best illustrated through an example. Suppose that $\omega$ is the permutation of the set of elements $\{1, 2, 3, 4\}$ that swaps 1 and 2 and 3 and 4. Then in one-line notation we would write $\omega = 2\ 1\ 4\ 3$. 2 is in the first position since 1 is sent to 2, 1 is in the second position since 2 is sent to 1, 4 is in the third position since 3 is sent to 4, and 3 is in the fourth position since 4 is sent to 3.

Permutations can be written as sequences of transpositions of adjacent elements. For instance, the permutation $\sigma = 3\ 1\ 2$ can be formed by swapping $1\ 2\ 3 \rightarrow 1\ 3\ 2 \rightarrow 3\ 1\ 2$. If we denote a transposition of the $i$th and $(i + 1)$st element as $s_i$ and read from right to left (as is the convention) then $\sigma$ can be written as $s_1 s_2$. A sequence of adjacent transpositions $s_{i_1} s_{i_2} \ldots s_{i_k}$ corresponding to a permutation $\sigma$ is called a *reduced word* if there is no other representations that uses fewer than $k$ adjacent transpositions to represent $\sigma$. Finally, two reduced words are considered *commutation equivalent* if one can be obtained from another by swaps of the form $s_i s_j \rightarrow s_j s_i$ where $|i - j| > 1$. A *descent* in a permutation $\sigma = a_1 a_2 \ldots a_n$ is a pair $(a_i, a_j)$ such that $i < j$ but $a_i > a_j$. The *descent set* of $\sigma$ is simply the set of all descents. A related notion is that of a $3412$ *pattern*. This is a quadruple $(a_i, a_j, a_k, a_\ell)$ such that $i < j < k < \ell$ but $a_k < a_\ell < a_i < a_j$. Descents and patterns have deep connections to algebra and geometry.

In the discussion above we implicitly thought of permutations of $n$ as bijective functions from $\{1, 2, \ldots, n\} \rightarrow \{1, 2, \ldots, n\}$. Using this perspective, one can define the composition of two permutations. The symmetric group, denoted $S_n$, is defined as the group of permutations on $n$ elements using composition as the group operation. (The sequence $s_1 s_2$ from the previous paragraph gave an example of the composition of two permutations).

**Posets:** A partially ordered set (poset) is a set $P$ of objects equipped with a binary relation, typically denoted "$\leq$", that is reflexive, antisymmetric, and transitive. This means that for all elements $a, b, c \in P$: (1) $a \leq a$, (2) If $a \leq b$ and $b \leq a$, then $b = a$, and (3) if $a \leq b$ and $b \leq c$, then $a \leq c$. Unlike total orders which are more familiar (e.g., $\mathbb{Z}$), in a partial order some pairs of elements may be incomparable. An example of a partially ordered set is the set of all subsets of $\{1, 2, 3, 4\}$, ordered by inclusion. This is a partial order and not a total order because $\{1, 2\}$ is not comparable to $\{2, 3\}$ or to $\{2, 3, 4\}$, for example. In a poset, $y$ covers $x$ if $y$ is greater than $x$ with respect to the ordering, and there is no element $z$ such that $y > z > x$. In this example, $\{1, 2, 4\}$ covers $\{1, 2\}$, $\{2, 4\}$, and $\{1, 4\}$, but not $\{1\}$, $\{2\}$, or $\{4\}$.

## C  Dataset details

All datasets are stored as either `.csv` or `.txt` files with one data instance per line. In this section we will describe each file and explain how to interpret it. We will make functions capable of loading and parsing each file available on our GitHub page.

We provide datasheet information that holds for (almost) all datasets in the collection here and provide answers that vary across each dataset in their relevant section.

- **Who funded the creation of the datasets?** Outside of the positive examples in Grassmannian cluster algebras dataset which were generated by an outside team, this research was supported by the Mathematics for Artificial Reasoning in Science (MARS) initiative at Pacific Northwest National Laboratory. It was conducted under the Laboratory Directed Research and Development (LDRD) Program at Pacific Northwest National Laboratory (PNNL), a multiprogram National Laboratory operated by Battelle Memorial Institute for the U.S. Department of Energy under Contract DE-AC05-76RL01830.

- **Are there recommended data splits (e.g., training, development/validation, testing)?** All datasets are provided in preset splits except the Grassmannian cluster algebras datasets, which draw on an external data source for positive examples. We will provide code to consistently split this data.

- **Are there any errors, sources of noise, or redundancies in the dataset?** Not that the creators are aware of.

- **Is the dataset self-contained, or does it link to or otherwise rely on external resources (e.g., websites, tweets, other datasets)?** Outside of the positive examples in the Grassmannian cluster algebras dataset, it is self contained.

- **Does the dataset contain data that might be considered confidential (e.g., data that is protected by legal privilege or by doctor– patient confidentiality, data that includes the content of individuals' non-public communications)?** No.

- **Does the dataset contain data that, if viewed directly, might be offensive, insulting, threatening, or might otherwise cause anxiety?** No.

- **Over what timeframe was the data collected?** All datasets (outside of the positive examples in the Grassmannian cluster algebra dataset) were generated in June 2024.

- **Were any ethical review processes conducted (e.g., by an institutional review board)?** N/A

- **Was the "raw" data saved in addition to the preprocessed/cleaned/labeled data (e.g., to support unanticipated future uses)?** Yes, Sage is open-source and freely available. We have made our Python and Sage code available. The code for generating the positive examples in the Grassmannian cluster algebras dataset is already available online, `https://github.com/edhirst/GrassmanniansML`.

- **Is the software that was used to preprocess/clean/label the data available?** Yes.

## C.1 Mutation equivalent quivers

### C.1.1 Context

Quivers and quiver mutations are central to the combinatorial study of cluster algebras, algebraic structures with connections to Poisson Geometry, string theory, and Teichmuller theory. Leaving precise definitions aside, quivers are directed graphs, and a quiver mutation is a local transformation of the graph involving certain vertices and arrows that produces a new quiver. A fundamental open problem in this area is to find an algorithm that determines whether two quivers are mutation equivalent. Currently, no such algorithm exists for quivers on more than three vertices.

Recent work has explored whether deep learning models can learn to correctly predict if two quivers are mutation equivalent [5]. Our dataset aims to facilitate continuation of this work. In [5] and in our dataset, quivers are represented using adjacency matrices.

**Dataset:** Adjacency matrices for seven quivers, each with 11 vertices, labeled by mutation equivalence class.

**Task:** Classify which mutation equivalence class an adjacency matrix corresponds to.

### C.1.2 The datasets themselves

The task associated with this dataset is identifying whether two quivers are mutation equivalent. Thus, the inputs are quivers (directed graphs) with 11 nodes that are encoded by their $11 \times 11$ adjacency matrices and the labels are one of 7 different equivalence classes: $A_{11}, BB_{11}, BD_{11}, BE_{11}, D_{11}, DE_{11}, E_{11}$. The files are organized by train and test for each of these classes. For the quiver mutation classes that are not mutation finite, the datasets contain quivers generated up to a certain depth, that is, the distance from the original quiver, measured by number of mutations. The depth is specified in the filename and was chosen to achieve as close to a balanced dataset as possible.

The file names are:

- `A_11_bmatrices_test.csv`
- `A_11_bmatrices_train.csv`
- `BB_11_depth10_bmatrices_test.csv`
- `BB_11_depth10_bmatrices_train.csv`

- `BD_11_depth9_bmatrices_test.csv`
- `BD_11_depth9_bmatrices_train.csv`
- `BE_11_depth8_bmatrices_test.csv`
- `BE_11_depth8_bmatrices_train.csv`
- `D_11_bmatrices_test.csv`
- `D_11_bmatrices_train.csv`
- `DE_11_depth9_bmatrices_test.csv`
- `DE_11_depth9_bmatrices_train.csv`
- `E_11_depth9_bmatrices_test.csv`
- `E_11_depth9_bmatrices_train.csv`

Within a file, each row is an adjacency matrix encoded in row major order.

### C.1.3 Mutation equivalent quivers datasheet

- **For what purpose was the dataset created?** To study the problem of determining whether two quivers are mutation equivalent.

- **Who created the dataset (e.g., which team, research group) and on behalf of which entity (e.g., company, institution, organization)?** The dataset was created by Helen Jenne at Pacific Northwest National Laboratory.

- **What do the instances that comprise the dataset represent (e.g., documents, photos, people, countries)?** The instances are $11 \times 11$ adjacency matrices corresponding to 11 vertex quivers.

- **How many instances are there in total (of each type, if appropriate)?** The number of instances of each type are described in the following table:

|            | Train  | Test  |
|------------|--------|-------|
| $AA_{11}$  | 11,940 | 2,984 |
| $BB_{11}$  | 27,410 | 6,852 |
| $BD_{11}$  | 23,651 | 5,912 |
| $BE_{11}$  | 22,615 | 5,653 |
| $D_{11}$   | 25,653 | 6,413 |
| $DE_{11}$  | 23,528 | 5,881 |
| $E_{11}$   | 28,998 | 7,249 |

- **Does the dataset contain all possible instances or is it a sample (not necessarily random) of instances from a larger set?** For the mutation classes $A$ and $D$, the dataset contains all possible instances. The other mutation classes are not finite, so it is not possible to generate all possible instances. For these mutation classes, we specify a depth $d$ (the number of mutations away from the original quiver) and the dataset contains all quivers at most $d$ mutations away from the original quiver.

- **What data does each instance consist of?** Each instance consists of an $11 \times 11$ adjacency matrix which represents an 11 vertex quiver.

- **Is there a label or target associated with each instance?** Yes, the label is specified in the file name in which the instance is stored.

- **Is any information missing from individual instances?** No.

- **Are relationships between individual instances made explicit (e.g., users' movie ratings, social network links)?** Yes

- **How was the data associated with each instance acquired?** The creators used the open-source mathematics software system SageMath to generate all quivers from each mutation class up to the specified depth. (See `https://doc.sagemath.org/html/en/reference/combinat/sage/combinat/cluster_algebra_quiver/quiver_mutation_type.html` for the relevant documentation).

- **What mechanisms or procedures were used to collect the data (e.g., hardware apparatuses or sensors, manual human curation, software programs, software APIs)?** SageMath was used for the calculations and Python was used to sort, format, and split the data.

- **If the dataset is a sample from a larger set, what was the sampling strategy (e.g., deterministic, probabilistic with specific sampling probabilities)?** Sampling was done by applying 9 uniform, randomly sampled mutations of the quiver in non-finite type cases. For finite type cases all examples were included.

- **Who was involved in the data collection process (e.g., students, crowdworkers, contractors) and how were they compensated (e.g., how much were crowdworkers paid)?** Helen Jenne used code from SageMath to generate this dataset. The authors of the SageMath QuiverMutationType class are Gregg Musiker and Christian Stump, and Hugh Thomas.

- **Was any preprocessing/cleaning/labeling of the data done (e.g., discretization or bucketing, tokenization, part-of-speech tagging, SIFT feature extraction, removal of instances, processing of missing values)?** Graphs were converted to adjacency matrices and then flattened for storage.

- **Has the dataset been used for any tasks already?** A similar dataset was used in [5] to study the ability of Naive Bayes and convolutional neural networks to classify quivers according to mutation class. Their dataset also included type $A$, $D$, and $E$ mutation classes, but for smaller values of $N$ (the number of vertices) and consequently, fewer instances of each class. [5] inspired this dataset, as it seemed interesting to investigate whether better classification performance could be achieved with larger values of $N$.

- **Is there a repository that links to any or all papers or systems that use the dataset?** Papers that use the dataset will be listed at `https://github.com/pnnl/ML4AlgComb`.

- **What (other) tasks could the dataset be used for?** This dataset could be used for any tasks focused on matching quivers to their mutation equivalence class.

- **Is there anything about the composition of the dataset or the way it was collected and preprocessed/cleaned/labeled that might impact future uses?** No.

- **Are there tasks for which the dataset should not be used? If so, please provide a description.** No.

- **Will the dataset be distributed to third parties outside of the entity (e.g., company, institution, organization) on behalf of which the dataset was created? If so, please provide a description.** No.

- **How will the dataset will be distributed (e.g., tarball on website, API, GitHub)? Does the dataset have a digital object identifier (DOI)?** All datasets will be compressed to a single `.zip` file and stored on Google Drive. Digital object identifiers will be generated before the paper and datasets are publicly announced.

- **Will the dataset be distributed under a copyright or other intellectual property (IP) license, and/or under applicable terms of use (ToU)?** CC0, `https://creativecommons.org/public-domain/cc0/`

- **Have any third parties imposed IP-based or other restrictions on the data associated with the instances?** No.

### C.2 Semistandard Young tableau for the Grassmannian

#### C.2.1 Context

The Grassmann manifold $\mathrm{Gr}(k, n)$ is the set of full-rank $k \times n$ matrices up to equivalence of elementary row operations (equivalently the space whose points are $k$-dimensional subspaces in $\mathbb{R}^n$). Grassmannians are of fundamental geometric importance and are a central tool in a model of quantum field theory known as supersymmetric Yang-Mills theory.

Among the many algebraic-combinatorial properties of Grassmannians is an algebraic structure on its coordinate ring making it a cluster algebra. A recent result of Chang, Duan, Fraser, and Li [8] parameterize cluster variables of the Grassmannian coordinate ring in terms of equivalence classes of semistandard Young tableaux. Not every semistandard Young tableaux indexes a cluster variable and

a natural question to ask is which are valid cluster variable indices. A necessary condition is that the tableaux is of rectangular shape. We follow the set-up of [10] who first applied machine learning to this problem, though we choose a different method of sampling tableau that do not index cluster variables.

**Dataset:** A collection of rectangular semistandard Young tableau each with a label indicating whether they index a cluster variable or not.

**Task:** Predict whether a Young tableaux indexes a cluster variable.

### C.2.2 The datasets themselves

This task is determining whether a semistandard Young tableaux corresponds to a cluster variable or not. The dataset we provide consists of tableau of shape $3 \times 4$, filled with values from $1, 2, \ldots, 12$. The files we provide are:

- `3_4_12_invalid_train.txt`
- `3_4_12_invalid_test.txt`

These contain tableau that do not correspond to cluster variables, 'invalid examples'.

We use braces $[$ and $]$ to demarcate rows of the dataset, so that

$$[[2,\ 3,\ 4,\ 7],\ [3,\ 5,\ 6,\ 8],\ [4,\ 9,\ 11,\ 12]]$$

corresponds to the tableau pictured in Figure 5

| 2 | 3 | 4 | 7 |
|---|---|---|---|
| 3 | 5 | 6 | 8 |
| 6 | 9 | 11 | 12 |

Figure 5: An example of a tableau from the Grassmannian cluster algebra dataset.

### C.2.3 SSYT for Grassmannian datasheet

- **For what purpose was the dataset created?** This dataset was created to study a machine learning model's ability to identify whether an SSYT indexes a valid cluster variable in the Grassmannian cluster algebra.
- **Who created the dataset (e.g., which team, research group) and on behalf of which entity (e.g., company, institution, organization)?** The positive dataset was created by an external team consisting of Man-Wai Cheung, Pierre-Phillips Dechant, Yang-Hui He, Elli Heyes, Edward Hirst, and Jian-Rong Li [10].
- **What do the instances that comprise the dataset represent (e.g., documents, photos, people, countries)?** Instances represent semistandard young tableau.
- **How many instances are there in total (of each type, if appropriate)?** There are $92,911$ instances of valid SSYT and $92,911$ instances of invalid SSYT.
- **Does the dataset contain all possible instances or is it a sample (not necessarily random) of instances from a larger set?** The positive instances are all possible instances with exceedingly high probability. An equal number of negative instances were obtained by randomly sampling without duplicates.
- **What data does each instance consist of?** An instance consists of a $3 \times 4$ SSYT.
- **Is there a label or target associated with each instance?** The label is given in the filename.
- **Is any information missing from individual instances?** No.
- **Are relationships between individual instances made explicit (e.g., users' movie ratings, social network links)?** Individual instances are unrelated.

- **How was the data associated with each instance acquired?** The creators generated the data using Python scripts.

- **What mechanisms or procedures were used to collect the data (e.g., hardware apparatuses or sensors, manual human curation, software programs, software APIs)?** Python was used to generate the data.

- **If the dataset is a sample from a larger set, what was the sampling strategy (e.g., deterministic, probabilistic with specific sampling probabilities)?** Positive examples are generated probabilistically until no new instances are generated after a large number of iterations. Negative examples are sampled uniformly at random.

- **Who was involved in the data collection process (e.g., students, crowdworkers, contractors) and how were they compensated (e.g., how much were crowdworkers paid)?** The positive dataset was created by an external team consisting of Man-Wai Cheung, Pierre-Phillips Dechant, Yang-Hui He, Elli Heyes, Edward Hirst, and Jian-Rong Li [10].

- **Was any preprocessing/cleaning/labeling of the data done (e.g., discretization or bucketing, tokenization, part-of-speech tagging, SIFT feature extraction, removal of instances, processing of missing values)?** Semistand Young tableau were flattened to fit on a single line in the data files.

- **Has the dataset been used for any tasks already?** A similar dataset was used in [10], with different negative examples.

- **Is there a repository that links to any or all papers or systems that use the dataset?** Papers that use the dataset will be listed at `https://github.com/pnnl/ML4AlgComb`.

- **What (other) tasks could the dataset be used for?** This dataset could be used for any tasks around the study of the Grassmannian cluster algebra.

- **Is there anything about the composition of the dataset or the way it was collected and preprocessed/cleaned/labeled that might impact future uses?** No.

- **Are there tasks for which the dataset should not be used? If so, please provide a description.** No.

- **Will the dataset be distributed to third parties outside of the entity (e.g., company, institution, organization) on behalf of which the dataset was created? If so, please provide a description.** No.

- **How will the dataset will be distributed (e.g., tarball on website, API, GitHub)? Does the dataset have a digital object identifier (DOI)?** The invalid examples that we generated will be compressed into a single `.zip` file and stored on Google Drive. The negative examples can be found at `https://github.com/edhirst/GrassmanniansML`. We will provide code to process these into preset splits. Digital object identifiers will be generated before the paper and datasets are publicly announced.

- **Will the dataset be distributed under a copyright or other intellectual property (IP) license, and/or under applicable terms of use (ToU)?** CC0, `https://creativecommons.org/public-domain/cc0/`

- **Have any third parties imposed IP-based or other restrictions on the data associated with the instances?** No.

### C.3 Kazhdan-Lusztig polynomials

#### C.3.1 Context

Kazhdan-Lusztig (KL) polynomials are integer polynomials in a variable $q$ that (for our purposes) are indexed by a pair of permutations [19]. We will write the KL polynomial associated with permutations $\sigma$ and $\nu$ as $P_{\sigma,\nu}(q)$. For example, the KL polynomial associated with permutations $\sigma = 1\ 4\ 3\ 2\ 7\ 6\ 5\ 10\ 9\ 8\ 11$ and $\nu = 4\ 6\ 7\ 8\ 9\ 10\ 1\ 11\ 2\ 3\ 5$ is

$$P_{\sigma,\nu} = 1 + 16q + 103q^2 + 337q^3 + 566q^4 + 529q^5 + 275q^6 + 66q^7 + 3q^8$$

(this example comes from [1]). KL polynomials have deep connections throughout several areas of mathematics. For example, KL polynomials are related to the dimensions of intersection homology in Schubert calculus, the study of the Hecke algebra, and representation theory of the symmetric

group. They can be computed via a recursive formula [19], nevertheless, in many ways they remain mysterious. For instance, there is no known closed formula for the degree of $P_{\sigma,\nu}(q)$.

One type of question of special interest is the value of coefficient on the terms of $q$ in $P_{\sigma,\nu}(q)$. Perhaps most well-known is the question of the coefficient on term $q^{\ell(\sigma)-\ell(\nu)-1/2}$ (where $\ell(x)$ is a statistic called the length of the permutation), which is known as the $\mu$-coefficient. Better understanding of this and other coefficients has the potential to shed considerable light on other aspects of this family of polynomials.

**Dataset:** Each instance in this dataset consists of a pair of permutations on $n$, $\sigma$ and $\nu$, along with the coefficients of $P_{\sigma,\nu}(q)$. We provide $n = 8, 9, 10$.

**Task:** The task to predict the coefficients of $P_{\sigma,\nu}(q)$ given $\sigma$ and $\nu$.

### C.3.2 The datasets themselves

Kazhdan-Lusztig polynomials are integer polynomials in a variable $q$ which are parametrized by two permutations. The task associated with this dataset is to predict the coefficients of the polynomial. Thus the input is two permutations $v, w$ and the output is a sequence of integers which are the coefficients of the polynomial. For instance, if $v = 0\ 2\ 1\ 3\ 5\ 4\ 6\ 9\ 7\ 8$ and $w = 2\ 3\ 0\ 5\ 9\ 6\ 7\ 8\ 1\ 4$ then

$$P_{v,w}(q) = 4q^4 + 12q^3 + 13q^2 + 6q + 1.$$

and this is written as the line

```
0213546978 2305967814 1,6,13,12,4.
```

The files we provide are:

- `processed_S5_train.txt`
- `processed_S6_test.txt`

### C.3.3 Kazhdan-Lusztig polynomial datasheet

- **For what purpose was the dataset created?** This dataset was created to study a machine learning model's ability to predict the coefficients of Kazhdan-Lusztig polynomials.

- **Who created the dataset (e.g., which team, research group) and on behalf of which entity (e.g., company, institution, organization)?** The dataset was created by Herman Chau at the University of Washington.

- **What do the instances that comprise the dataset represent (e.g., documents, photos, people, countries)?** Instances represent a Kazhdan-Lusztig polynomial (the two permutations that index it and its coefficients up to the largest non-zero one).

- **How many instances are there in total (of each type, if appropriate)?** There are $98,407$ instances total in the dataset for $n = 6$.

- **Does the dataset contain all possible instances or is it a sample (not necessarily random) of instances from a larger set?** Yes, all non-zero instances in $S_6$ are included.

- **What data does each instance consist of?** An instance consists of a pair of permutations followed by the Kazhdan-Lusztig polynomial corresponding to the pair of permutations.

- **Is there a label or target associated with each instance?** The labels can be taken to be the coefficients, the degree, etc.

- **Is any information missing from individual instances?** No.

- **Are relationships between individual instances made explicit (e.g., users' movie ratings, social network links)?** Individual instances are unrelated.

- **Are there recommended data splits (e.g., training, development/validation, testing)?** We provide the data in preset split files.

- **How was the data associated with each instance acquired?** The creators generated the data using SageMath and Python scripts.

- **What mechanisms or procedures were used to collect the data (e.g., hardware apparatuses or sensors, manual human curation, software programs, software APIs)?** SageMath and Python was used to generate the data.

- **If the dataset is a sample from a larger set, what was the sampling strategy (e.g., deterministic, probabilistic with specific sampling probabilities)?** N/A.

- **Who was involved in the data collection process (e.g., students, crowdworkers, contractors) and how were they compensated (e.g., how much were crowdworkers paid)?** Herman Chau wrote SageMath and Python code to generate the data.

- **Was any preprocessing/cleaning/labeling of the data done (e.g., discretization or bucketing, tokenization, part-of-speech tagging, SIFT feature extraction, removal of instances, processing of missing values)?** Polynomials were stored as a sequence of coefficients ending after the final non-zero coefficient.

- **Has the dataset been used for any tasks already?** No.

- **Is there a repository that links to any or all papers or systems that use the dataset?** Papers that use the dataset will be listed at `https://github.com/pnnl/ML4AlgComb`.

- **What (other) tasks could the dataset be used for?** This dataset could be used to study various properties of the coefficients of Kazhdan-Lusztig polynomials.

- **Is there anything about the composition of the dataset or the way it was collected and preprocessed/cleaned/labeled that might impact future uses?** No.

- **Are there tasks for which the dataset should not be used? If so, please provide a description.** No.

- **Will the dataset be distributed to third parties outside of the entity (e.g., company, institution, organization) on behalf of which the dataset was created? If so, please provide a description.** No.

- **How will the dataset will be distributed (e.g., tarball on website, API, GitHub)? Does the dataset have a digital object identifier (DOI)?** The dataset will be compressed to a single `.zip` file and stored on Google Drive. Digital object identifiers will be generated before the paper and datasets are publicly announced.

- **Will the dataset be distributed under a copyright or other intellectual property (IP) license, and/or under applicable terms of use (ToU)?** CC0, `https://creativecommons.org/public-domain/cc0/`

- **Have any third parties imposed IP-based or other restrictions on the data associated with the instances?** No.

### C.4 Partial orders on lattice paths:

#### C.4.1 Context

[25] defines two order relations on NE lattice paths from $(0,0)$ to $(a,b)$ called the *matching ordering* ($\leq_M$) and the *Lagrange ordering* ($\leq_L$), and proposes studying these partially ordered sets. The matching ordering assigns a number to each lattice path based on the number of perfect matchings of an associated snake graph, while the Lagrange ordering assigns a number to each lattice path equal to the Lagrange number of a continued fraction. These numbers each define the respective partial order. An open question related to the matching and Lagrange orders is whether we can find a simple way of determining whether two paths $w$ and $w'$ have the same relationship in both orders ($w \leq_L w'$ and $w \leq_M w'$) or different relationships in both orders ($w \leq_L w'$ and $w \geq_M w'$ or vice versa) [3].

**Dataset:** Pairs of NE lattice paths $(w, w')$ on a grid of size $n \times n - 1$ where $w'$ covers $w$ in either the matching or Lagrange order (but not both). We include $n = 10, 11, 12, 13$.

**Task (classification):** Train a model that can predict whether $(w, w')$ is a covering pair in Lagrange or matching order.

**Dataset:** Each instance in this dataset consists of a pair of permutations on $n$, $\sigma$ and $\nu$, along with the coefficients of $P_{\sigma,\nu}(q)$. We provide $n = 8, 9, 10$.

**Task:** The task to predict the coefficients of $P_{\sigma,\nu}(q)$ given $\sigma$ and $\nu$.

### C.4.2 The datasets themselves

This dataset contains pairs of lattice paths starting at $(0, 0)$ and ending at $(n, n-1)$ that are only allowed to take steps either north or east, and stay below the line $y = \frac{n}{n-1}x$. They are thus encoded by a sequence of 1's (for steps east) and 0's (for steps north) of length $(n+1) + n = 2n + 1$. Each pair of lattice paths is a covering pair in exactly one of the two partial orders, the Lagrange order or the matching order. The task is to predict which partial order it is a cover in.

Each line in a file is the concatenation of two 0-1 sequences (one for each path) for a length $4n + 2$ row of 0's and 1's. The lattice paths are separated by ';'.

So for an $3 \times 2$ grid, the sequence:

$$1, 1, 1, 0, 0; 1, 1, 0, 1, 0$$

corresponds to the two lattice paths in Figure 6. The first is in red and second is in blue, with segments traversed by both paths colored red. This small example is meant to serve as an example of how to translate between sequences and pictures and does not correspond to an actual path with its cover in either ordering.

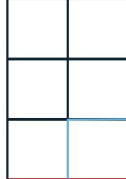

Figure 6: An example of two lattice paths from $(0, 0)$ to $(3, 2)$. These do not correspond to covers.

We store files for $n = 10, 11, 12$. These are named:

- `lagrange_covers_test_10_9.csv`
- `lagrange_covers_test_11_10.csv`
- `lagrange_covers_test_12_11.csv`
- `lagrange_covers_test_13_12.csv`
- `lagrange_covers_train_10_9.csv`
- `lagrange_covers_train_11_10.csv`
- `lagrange_covers_train_12_11.csv`
- `lagrange_covers_train_13_12.csv`
- `matching_covers_test_10_9.csv`
- `matching_covers_test_11_10.csv`
- `matching_covers_test_12_11.csv`
- `matching_covers_test_13_12.csv`
- `matching_covers_train_10_9.csv`
- `matching_covers_train_11_10.csv`
- `matching_covers_train_12_11.csv`
- `matching_covers_train_13_12.csv`

The first word ('Lagrange' or 'matching') gives the label, the third word says whether it is train or test, and the final two numbers give $n$ and $n - 1$.

### C.4.3 Partial orders on lattice paths datasheet

- **For what purpose was the dataset created?** This dataset was created to study a machine learning model's ability to differentiate the Lagrange and matching partial orders.

- **Who created the dataset (e.g., which team, research group) and on behalf of which entity (e.g., company, institution, organization)?** The dataset was created by Helen Jenne at Pacific Northwest National Laboratory.

- **What do the instances that comprise the dataset represent (e.g., documents, photos, people, countries)?** Instances represent two lattice paths $(p, q)$, where $q$ is a cover of $p$ in either the Lagrange ordering or the matching ordering.

- **How many instances are there in total (of each type, if appropriate)?** The instances of each type are described in the following table:

| | 10 | 11 | 12 | 13 |
|---|---|---|---|---|
| Lagrange | 9,453 | 33,028 | 116,542 | 413,854 |
| matching | 4,843 | 16,779 | 58,725 | 207,884 |

- **Does the dataset contain all possible instances or is it a sample (not necessarily random) of instances from a larger set?**

  The dataset contains the vast majority of possible instances, but pairs $(p, q)$ that were covering pairs in both the Lagrange partial order and the matching partial order were thrown out (this was 21, 40, 79, and 183 instances for $n = 10, 11, 12$, and 13, respectively)

- **What data does each instance consist of?** An instance in the dataset consists of two lattice paths represented as binary sequences.

- **Is there a label or target associated with each instance?** The label is given in the filename; the Lagrange and matching covers are saved in separate files.

- **Is any information missing from individual instances?** No.

- **Are relationships between individual instances made explicit (e.g., users' movie ratings, social network links)?**

  Each lattice path does not necessarily have a unique cover, so there are instances in the dataset that have the same first lattice path. This is the reason for the dataset imbalance: lattice paths have unique covers less often in the Lagrange partial ordering.

- **How was the data associated with each instance acquired?** The creators used the open-source mathematics software system SageMath to generate all lattice paths from $(0, 0)$ to $(n + 1, n)$ that stay below the diagonal $y = \frac{n+1}{n}x$, and compute the Lagrange number $L(p)$ and matching number $M(p)$ associated to each lattice path $p$. The matching order (resp. Lagrange order) dataset consists of lattice paths $(p, q)$ such that $M(q) > M(p)$ (resp. $L(q) > L(p)$) and there is not a path $r$ such that $M(q) > M(r) > M(p)$ (resp. $L(q) > L(r) > L(p)$).

- **What mechanisms or procedures were used to collect the data (e.g., hardware apparatuses or sensors, manual human curation, software programs, software APIs)?** SageMath was used for the calculations and Python was used to sort, format, and split the data.

- **If the dataset is a sample from a larger set, what was the sampling strategy (e.g., deterministic, probabilistic with specific sampling probabilities)?** The dataset is not a sample from a larger dataset, but we did throw out instances that were covering pairs in both the Lagrange partial order and the matching partial order (this was 21, 40, 79, and 183 instances for $n = 10, 11, 12$, and 13, respectively)

- **Who was involved in the data collection process (e.g., students, crowdworkers, contractors) and how were they compensated (e.g., how much were crowdworkers paid)?** Helen Jenne wrote the code in SageMath to generate this dataset.

- **Was any preprocessing/cleaning/labeling of the data done (e.g., discretization or bucketing, tokenization, part-of-speech tagging, SIFT feature extraction, removal of instances, processing of missing values)?** We converted lattice paths to binary codes of 0's and 1's for storage.

- **Has the dataset been used for any tasks already?** The Lagrange and matching orderings are an area of recent interest in the algebraic combinatorics community, but this specific dataset has never been used before.

- **Is there a repository that links to any or all papers or systems that use the dataset?** Papers that use the dataset will be listed at `https://github.com/pnnl/ML4AlgComb`.

- **What (other) tasks could the dataset be used for?** This dataset could be used for any tasks around the study of the Lagrange and matching orderings.
- **Is there anything about the composition of the dataset or the way it was collected and preprocessed/cleaned/labeled that might impact future uses?** No.
- **Are there tasks for which the dataset should not be used? If so, please provide a description.** No.
- **Will the dataset be distributed to third parties outside of the entity (e.g., company, institution, organization) on behalf of which the dataset was created? If so, please provide a description.** No.
- **How will the dataset will be distributed (e.g., tarball on website, API, GitHub)? Does the dataset have a digital object identifier (DOI)?** All datasets will be compressed to a single `.zip` file and stored on Google Drive. Digital object identifiers will be generated before the paper and datasets are publicly announced.
- **Will the dataset be distributed under a copyright or other intellectual property (IP) license, and/or under applicable terms of use (ToU)?** CC0, `https://creativecommons.org/public-domain/cc0/`
- **Have any third parties imposed IP-based or other restrictions on the data associated with the instances?** No.

### C.5 The mHeight function

#### C.5.1 Context

The mHeight function is a statistic associated with a permutation that relates to all $3412$-patterns in the permutation (see Section B for the definition of a $3412$-pattern). It plays a crucial role in the proof by Gaetz and Gao [14] which resolved a long-standing conjecture of Billey and Postnikov [7] about the coefficients on Kazhdan-Lusztig polynomials which carry important geometric information about certain spaces, Schubert varieties, that are of interest both to mathematicians and physicists. The task of predicting the mHeight function thus represents an interesting opportunity to understand whether a non-trivial intermediate step in an important proof can be learned by machine learning.

Let $\sigma = a_1 \ldots a_n \in S_n$ be a permutation containing at least one occurrence of a $3412$ pattern. Let $(a_i, a_j, a_k, a_\ell)$ be a $3412$ pattern so that $1 \leq i < j < k < \ell \leq n$ but $a_k < a_\ell < a_i < a_j$. The *height* of $(a_i, a_j, a_k, a_\ell)$ is $a_\ell - a_i$ (see Figure 3 in the Appendix for an example). The *mHeight* of $\sigma$ is then the minimum height over all $3412$ patterns in $\sigma$.

**Dataset:** Permutations of size $n$ labeled by their mHeight. We provide datasets for $n = 10, 11, 12$.

**Task:** Predict the mHeight of a permutation.

#### C.5.2 The datasets themselves

The mHeight function dataset contains permutations which are labeled by their height. Unlike some of the other datasets where permutations are written in one-line notation, in this dataset we write them in terms of their inversion set. For a permutation $\sigma$ on $n$ elements, the inversion set gives all pairs of numbers $1 \leq i < j \leq n$ such that $\sigma(j) < \sigma(i)$. There are $\binom{n}{2}$ possible inversions for $\sigma$. We provide a binary code where 1 means that $\sigma$ inverts $(i, j)$ and 0 means that it does not. Note that an inversion set completely characterizes a permutation.

We list permutations in lexicographical order so that the full list of inversions of $\{1, 2, 3\}$ is

$$(1, 2), \ (1, 3), \ (2, 3).$$

The permutation 213, inverts $(1, 2)$ but not $(1, 3)$ or $(2, 3)$, so we would write it as $1, 0, 0$.

The mheight function value (which is a function of its inversion set) is written after the ';' character.

We store files for $n = 10, 11, 12$. These are named:

- `mHeight_10_train.txt`
- `mHeight_10_test.txt`

- `mHeight_11_train.txt`
- `mHeight_11_test.txt`
- `mHeight_12_train.txt`
- `mHeight_12_test.txt`

### C.5.3 mHeight datasheet

- **For what purpose was the dataset created?** This dataset was created to study a machine learning model's ability to learn the mHeight of a permutation.

- **Who created the dataset (e.g., which team, research group) and on behalf of which entity (e.g., company, institution, organization)?** The dataset was created by Herman Chau at the University of Washington.

- **What do the instances that comprise the dataset represent (e.g., documents, photos, people, countries)?** Instances are permutations (represented in one-line notation) followed by the integer corresponding to their mHeight.

- **How many instances are there in total (of each type, if appropriate)?**

|       | 10      | 11        | 12         |
|-------|---------|-----------|------------|
| Train | 374,112 | 2,627,257 | 18,464,505 |
| Test  | 93,528  | 656,815   | 4,616,127  |

- **Does the dataset contain all possible instances or is it a sample (not necessarily random) of instances from a larger set?** Yes, all instances are included.

- **What data does each instance consist of?** An instance consists of a permutation and its corresponding mHeight.

- **Is there a label or target associated with each instance?** The label is the mHeight.

- **Is any information missing from individual instances?** No.

- **Are relationships between individual instances made explicit (e.g., users' movie ratings, social network links)?** Individual instances are unrelated.

- **How was the data associated with each instance acquired?** The creators generated the data using Python scripts.

- **What mechanisms or procedures were used to collect the data (e.g., hardware apparatuses or sensors, manual human curation, software programs, software APIs)?** Python was used to generate the data.

- **If the dataset is a sample from a larger set, what was the sampling strategy (e.g., deterministic, probabilistic with specific sampling probabilities)?** N/A

- **Who was involved in the data collection process (e.g., students, crowdworkers, contractors) and how were they compensated (e.g., how much were crowdworkers paid)?** Herman Chau wrote Python code to generate the data.

- **Was any preprocessing/cleaning/labeling of the data done (e.g., discretization or bucketing, tokenization, part-of-speech tagging, SIFT feature extraction, removal of instances, processing of missing values)?** No.

- **Has the dataset been used for any tasks already?** No.

- **Is there a repository that links to any or all papers or systems that use the dataset?** Papers that use the dataset will be listed at `https://github.com/pnnl/ML4AlgComb`.

- **What (other) tasks could the dataset be used for?** This dataset could be used to predict the mHeight of permutations and as an intermediate task for predicting tho smallest non-trivial zero coefficient of Kazhdan-Lusztig polynomials.

- **Is there anything about the composition of the dataset or the way it was collected and preprocessed/cleaned/labeled that might impact future uses?** No.

- **Are there tasks for which the dataset should not be used? If so, please provide a description.** No.

- **Will the dataset be distributed to third parties outside of the entity (e.g., company, institution, organization) on behalf of which the dataset was created? If so, please provide a description.** No.

- **How will the dataset will be distributed (e.g., tarball on website, API, GitHub)? Does the dataset have a digital object identifier (DOI)?** All datasets will be compressed to a single `.zip` file and stored on Google Drive. Digital object identifiers will be generated before the paper and datasets are publicly announced.

- **Will the dataset be distributed under a copyright or other intellectual property (IP) license, and/or under applicable terms of use (ToU)?** CC0, `https://creativecommons.org/public-domain/cc0/`

- **Have any third parties imposed IP-based or other restrictions on the data associated with the instances?** No.

### C.6 The Robinson–Schensted-Knuth correspondence

#### C.6.1 Context

The goal of this benchmark is to see whether a model can learn the RSK algorithm. That is, for a fixed $n$ the model is provided with a permutation $\pi \in S_n$ and required to predict pairs of standard Young tableaux. Although the algorithm is known, it would be significant for a model to learn this correspondence due to the the intricate combinatorial rules involved. Notably, the RSK correspondence can be used to find the length of the longest increasing subsequence, so a model that learns this algorithm implicitly must also learn to solve the increasing subsequence problem. Additionally, given the numerous generalizations of the RSK correspondence, a model that performs well on this benchmark could potentially be investigated for its ability to generalize to other related combinatorial settings.

**Dataset:** This dataset consists of triples: two standard Young tableau of size $n$ and their corresponding permutation (via the RSK algorithm). We include datasets for $n = 8, 9, 10$.

**Task:** Given pairs of standard Young tableau, predict the corresponding permutation.

#### C.6.2 The datasets themselves

The Robinson–Schensted-Knuth correspondence assigns to every permutation of $n$ a pair of standard Young tableau of the same shape (that is, the shape corresponds to the conjugacy class of $n$). This dataset thus consists of files of permutations of $n$ for input and files with pairs of standard Young tableau as output. For instance,

We store files for $n = 8, 9, 10$. These are named:

- `input_permutations_8_train.csv`
- `input_permutations_8_test.csv`
- `input_permutations_9_train.csv`
- `input_permutations_9_test.csv`
- `input_permutations_10_train.csv`
- `input_permutations_10_test.csv`
- `output_tableau_8_train.csv`
- `output_tableau_8_test.csv`
- `output_tableau_9_train.csv`
- `output_tableau_9_test.csv`
- `output_tableau_10_train.csv`
- `output_tableau_10_test.csv`

The input permutations are stored in files starting with `input_permutation` and the pair of output tableau are stored in files labeled by `output_tableau`. Permutations are stored in 1-line notation while tableau rows are separated by '[' and ']'. For instance,

```
[[[1, 3, 4], [2, 7], [5], [6]], [[1, 2, 6], [3, 4], [5], [7]]]
```
corresponds to the pair of Young tableau in Figure 7.

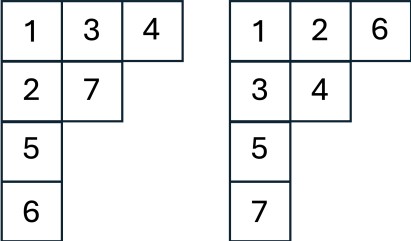

Figure 7: The Young tableau pair that are obtained by applying the Robinson-Schensted-Knuth algorithm to the permutation 6 7 2 5 3 4 1.

The Robinson-Schensted correspondence is easily reversed, so for our initial baseline model we built a transformer that takes as input a pair of standard Young tableau and outputs a permutation as its inversion vector (a binary sequence). We envision the dataset being used to predict both directions.

### C.6.3 Robinson-Schensted datasheet

- **For what purpose was the dataset created?** This dataset was created to study a machine learning model's ability to learn the Robinson-Schensted correspondence.

- **Who created the dataset (e.g., which team, research group) and on behalf of which entity (e.g., company, institution, organization)?** The dataset was created by Henry Kvinge and Helen Jenne at Pacific Northwest National Laboratory.

- **What do the instances that comprise the dataset represent (e.g., documents, photos, people, countries)?**
  Each instance in the dataset is a pair of Standard Young Tableaux of the same shape, along with their associated permutation under the Robinson-Schensted-Knuth correspondence.

- **How many instances are there in total (of each type, if appropriate)?**
  There are $n!$ permutations of $n$, so there are $40,320, 362,880$, and $3,628,800$ instances for $n = 8, 9$, and $10$, respectively.

- **Does the dataset contain all possible instances or is it a sample (not necessarily random) of instances from a larger set?**
  The dataset contains all possible instances for $n = 8, 9$, and $10$.

- **What data does each instance consist of?** An instance in the dataset consists of a pair of Standard Young Tableaux along with its associated permutation. .

- **Is there a label or target associated with each instance?** The target permutations are given in separate files.

- **Is any information missing from individual instances?** No.

- **Are relationships between individual instances made explicit (e.g., users' movie ratings, social network links)?** N/A

- **How was the data associated with each instance acquired?** The creators used the open-source mathematics software system SageMath to generate the tableaux pairs corresopnding to each permutation. (See `https://doc.sagemath.org/html/en/reference/combinat/sage/combinat/rsk.html` for the relevant documentation).

- **What mechanisms or procedures were used to collect the data (e.g., hardware apparatuses or sensors, manual human curation, software programs, software APIs)?** SageMath was used to generate the data and Python was used to sort, format, and split the data.

- **If the dataset is a sample from a larger set, what was the sampling strategy (e.g., deterministic, probabilistic with specific sampling probabilities)?** N/A

- **Who was involved in the data collection process (e.g., students, crowdworkers, contractors) and how were they compensated (e.g., how much were crowdworkers paid)?** Henry Kvinge and Helen Jenne used code from SageMath to generate this dataset. The authors of the original implementation of the Robinson-Schensted-Knuth correspondence in SageMath QuiverMutationType class is Travis Scrimshaw.

- **Was any preprocessing/cleaning/labeling of the data done (e.g., discretization or bucketing, tokenization, part-of-speech tagging, SIFT feature extraction, removal of instances, processing of missing values)?** Standard Young tableau were flattened to fit on a single line. Permutations were written as a list of numbers.

- **Has the dataset been used for any tasks already?** The Robinson-Schensted correspondence is an important algorithm in the algebraic combinatorics community, but this specific dataset has not been used before.

- **Is there a repository that links to any or all papers or systems that use the dataset?** Papers that use the dataset will be listed at `https://github.com/pnnl/ML4AlgComb`.

- **What (other) tasks could the dataset be used for?** This dataset could be used for any tasks involving the RSK algorithm.

- **Is there anything about the composition of the dataset or the way it was collected and preprocessed/cleaned/labeled that might impact future uses?** No.

- **Are there tasks for which the dataset should not be used? If so, please provide a description.** No.

- **Will the dataset be distributed to third parties outside of the entity (e.g., company, institution, organization) on behalf of which the dataset was created? If so, please provide a description.** No.

- **How will the dataset will be distributed (e.g., tarball on website, API, GitHub)? Does the dataset have a digital object identifier (DOI)?** All datasets will be compressed to a single `.zip` file and stored on Google Drive. Digital object identifiers will be generated before the paper and datasets are publicly announced.

- **Will the dataset be distributed under a copyright or other intellectual property (IP) license, and/or under applicable terms of use (ToU)?** CC0, `https://creativecommons.org/public-domain/cc0/`

- **Have any third parties imposed IP-based or other restrictions on the data associated with the instances?** No.

### C.7 Schubert polynomial structure coefficients

#### C.7.1 Context

When two Schubert polynomials are multiplied, their product is a linear combination of Schubert polynomials, i.e. $\mathfrak{S}_\beta \mathfrak{S}_\gamma = \sum_\alpha c_{\beta\gamma}^\alpha \mathfrak{S}_\alpha$. The question is whether the $c_{\beta\gamma}^\alpha$ (the structure constants) have a combinatorial description or formula. To give an example of what we mean by combinatorial description, in the case of Schur polynomials (which can be viewed as specific case of Schubert polynomials), the coefficients in the product are equal to the number of semistandard tableaux satisfying certain properties.

**Dataset:** Each instance in this dataset is a triple of permutations $(\beta, \gamma, \alpha)$, labeled by its coefficient $c_{\beta\gamma}^\alpha$ in the expansion of the product $\mathfrak{S}_\beta \mathfrak{S}_\gamma$. Not all possible triples of permutations are included; the dataset consists of an approximately equal number of zero and nonzero coefficients. We provide datasets for $n = 4, 5, 6$.

**Task:** The task is to predict the coefficient $c_{\beta\gamma}^\alpha$.

#### C.7.2 The datasets themselves

When a vector space also has a multiplicative structure so that we can multiply two vectors together, we can compute structure coefficients. For basis elements $v_\alpha, v_\beta, v_\gamma$, the structure constant $c_{\alpha,\beta}^\gamma$ is the

coefficients on a basis element $v_\gamma$ that we get when we multiply basis elements $v_\alpha$ and $v_\beta$ together.

$$v_\alpha \star v_\beta = \sum_{\gamma \in I} c_{\alpha,\beta}^\gamma v_\gamma.$$

This task involves predicting the structure constants of Schubert polynomials. These polynomials are indexed by permutations, $\{\mathcal{S}_v\}_{v \in S_\infty}$, hence, the input is a triple of permutations $v, w, u$ and the output is an integer. For instance we have:

$$\mathcal{S}_{12354}\mathcal{S}_{12354} = \mathcal{S}_{123645} + \mathcal{S}_{12453}.$$

Hence, one data instance is

```
[1,2,3,5,4],[1,2,3,5,4],[1,2,3,6,4,5];1].
```

We partition the datasets so that the dataset associated with value $n$ has structure constants for pairs of $\mathcal{S}_v$ with $v \in S_n$. Note that there is some repetition since $S_{n-1}$ is a subset of $S_n$. We store files for $n = 4, 5, 6$. These are named:

- `schubert_structure_coefficients_4_train.csv`
- `schubert_structure_coefficients_4_test.csv`
- `schubert_structure_coefficients_5_train.csv`
- `schubert_structure_coefficients_5_test.csv`
- `schubert_structure_coefficients_6_train.csv`
- `schubert_structure_coefficients_6_test.csv`.

### C.7.3 Schubert polynomial structure coefficients datasheet

- **For what purpose was the dataset created?** This dataset was created to study machine learning model's ability to predict the structure constants of Schubert polynomials.

- **Who created the dataset (e.g., which team, research group) and on behalf of which entity (e.g., company, institution, organization)?** The dataset was created by Henry Kvinge and Helen Jenne at Pacific Northwest National Laboratory.

- **What do the instances that comprise the dataset represent (e.g., documents, photos, people, countries)?** Instances represent the structure constants that come from multiplying Schubert polynomials together. For fixed $n$, instance

$$[\alpha, \beta, \gamma, c]$$

where $\alpha, \beta \in S_n$, $\gamma$ is another permutation (possibly in a larger or smaller symmetric group), and $c \in \mathbb{Z}_{\geq 0}$.

- **How many instances are there in total (of each type, if appropriate)?** For $n = 4$ there are 2105 instances, for $n = 5$ there are $107,025$ instances, for $n = 6$ there are $10,508,205$ instances. Each of these are split for $80\%$ train and $20\%$ test.

- **Does the dataset contain all possible instances or is it a sample (not necessarily random) of instances from a larger set?** For any $n$, most structure constants will be zero. To generate a balanced dataset, we computed $\mathcal{S}_\alpha \star \mathcal{S}_\beta$ for all elements in $S_n \times S_n$ and for each $c_{\alpha,\beta}^\gamma \neq 0$, we applied a transposition to $\gamma$ to get $\gamma'$, checked that $c_{\alpha,\beta}^{\gamma'} = 0$ and added this to the dataset. Therefore the dataset contains all non-zero structure constants but only a fraction of zero structure constants.

- **What data does each instance consist of?** An instance in the dataset corresponding to $n$ consists of two permutations from $S_n$, a permutation from another (possibly larger or smaller symmetric group), and an integer.

- **Is there a label or target associated with each instance?** The final integer in the instance is the label.

- **Is any information missing from individual instances?** No.

- **Are relationships between individual instances made explicit (e.g., users' movie ratings, social network links)?** Instances that have the same first two permutations $\alpha, \beta$ are drawn from the same basis expansion of $\mathcal{S}_\alpha \star \mathcal{S}_\beta$.

- **How was the data associated with each instance acquired?** The creators used the open-source mathematics software system SageMath to generate and multiple Schubert polynomials for each pair of permutations $\alpha$ and $\beta$ in $S_n$ for $n = 4, 5, 6$. The basis expansion of $\mathcal{S}_\alpha \star \mathcal{S}_\beta$ was obtained from this and each term in this expansion was used as an instance.

- **What mechanisms or procedures were used to collect the data (e.g., hardware apparatuses or sensors, manual human curation, software programs, software APIs)?** SageMath was used for the calculations and Python was used to sort, format, split the data.

- **If the dataset is a sample from a larger set, what was the sampling strategy (e.g., deterministic, probabilistic with specific sampling probabilities)?** The dataset contains all non-zero structure constants. To randomly choose a subset of zero structure constants to include we computed $\mathcal{S}_\alpha \star \mathcal{S}_\beta$ for all elements in $S_n \times S_n$ and for each $c_{\alpha,\beta}^\gamma \neq 0$, we applied a transposition to $\gamma$ to get $\gamma'$, checked that $c_{\alpha,\beta}^{\gamma'} = 0$ and added this to the dataset. This transposition was sampled from the uniform distribution over all transpositions over the support of the permutation $\gamma$.

- **Who was involved in the data collection process (e.g., students, crowdworkers, contractors) and how were they compensated (e.g., how much were crowdworkers paid)?** Henry Kvinge and Helen Jenne wrote the code in SageMath to generate this dataset.

- **Was any preprocessing/cleaning/labeling of the data done (e.g., discretization or bucketing, tokenization, part-of-speech tagging, SIFT feature extraction, removal of instances, processing of missing values)?** Permutations are represented as lists of integers enclosed by brackets.

- **Has the dataset been used for any tasks already?** Schubert structure constants are an area of intense interest to the algebraic combinatorics community, but this specific dataset has never been used before.

- **Is there a repository that links to any or all papers or systems that use the dataset?** Papers that use the dataset will be listed at `https://github.com/pnnl/ML4AlgComb`.

- **What (other) tasks could the dataset be used for?** This dataset could be used for any tasks around the study of Schubert polynomial structure constants.

- **Is there anything about the composition of the dataset or the way it was collected and preprocessed/cleaned/labeled that might impact future uses?** No.

- **Are there tasks for which the dataset should not be used? If so, please provide a description.** No.

- **Will the dataset be distributed to third parties outside of the entity (e.g., company, institution, organization) on behalf of which the dataset was created? If so, please provide a description.** No.

- **How will the dataset will be distributed (e.g., tarball on website, API, GitHub)? Does the dataset have a digital object identifier (DOI)?** All datasets will be compressed to a single `.zip` file and stored on Google Drive. Digital object identifiers will be generated before the paper and datasets are publicly announced.

- **Will the dataset be distributed under a copyright or other intellectual property (IP) license, and/or under applicable terms of use (ToU)?** CC0, `https://creativecommons.org/public-domain/cc0/`

- **Have any third parties imposed IP-based or other restrictions on the data associated with the instances?** No.

## C.8 Symmetric group characters

### C.8.1 Context

The representation theory of symmetric groups has rich combinatorial interpretations. Both irreducible representations of $S_n$ and the conjugacy classes of $S_n$ are indexed by partitions of $n$ and thus the

characters of irreducible representations of $S_n$ are indexed by two partitions of $n$. For $\lambda, \mu \vdash n$ we write $\chi_\mu^\lambda$. This combinatorial connection is not superficial, there are algorithms (e.g., the Murnaghan-Nakayama rule), which allow calculation of irreducible characters via simple manipulation of the Young diagrams for $\lambda$ and $\mu$ without any reference to algebra.

**Dataset/Task:** Pairs of integer partitions of $n$, $\lambda, \mu$ and the corresponding symmetric group character $\chi_\mu^\lambda$.

**Task:** Given partitions $\lambda$ and $\mu$, predict the irreducible symmetric group character $\chi_\mu^\lambda$.

### C.8.2 The datasets themselves

Since the conjugacy classes of the symmetric group $S_n$ are indexed by integer partitions of $n$, characters are constant on conjugacy classes, and the irreducible representations of $S_n$ are also indexed by integer partitions of $n$, this task takes in two integer partitions of $n$ and tries to predict the character of the corresponding irreducible representation of the symmetric group.

Within each file, two integer partitions are provided followed by an integer corresponding to the character. For instance:

```
[3,1,1],[2,2,1],-2
```

says that the character $\chi_{2\,2\,1}^{3\,1\,1} = -2$. We store files for $n = 4, 5, 6$. These are named:

- `sym_grp_char_18_train.txt`
- `sym_grp_char_18_test.txt`
- `sym_grp_char_20_train.txt`
- `sym_grp_char_20_test.txt`
- `sym_grp_char_22_train.txt`
- `sym_grp_char_22_test.txt`

### C.8.3 Symmetric group characters datasheet

- **For what purpose was the dataset created?** To study whether machine learning models can learn to predict the characters of irreducible representations of the symmetric group.

- **Who created the dataset (e.g., which team, research group) and on behalf of which entity (e.g., company, institution, organization)?** The dataset was created by Henry Kvinge at Pacific Northwest National Laboratory.

- **What do the instances that comprise the dataset represent (e.g., documents, photos, people, countries)?** Are two integer partitions of $n$ followed by an integer (the character).

- **How many instances are there in total (of each type, if appropriate)?** The size of each dataset is given in the following table:

|          | Train    | Test    |
|----------|----------|---------|
| $n = 18$ | 118,580  | 29,645  |
| $n = 20$ | 314,503  | 78,626  |
| $n = 22$ | 803,203  | 200,801 |

- **Does the dataset contain all possible instances or is it a sample (not necessarily random) of instances from a larger set?** It contains all possible instances for a fixed $n$.

- **What data does each instance consist of?** Each instance consists of two partitions of $n$ and an integer.

- **Is there a label or target associated with each instance?** Yes, the label is the final integer.

- **Is any information missing from individual instances?** No.

- **Are relationships between individual instances made explicit (e.g., users' movie ratings, social network links)?** N/A

- **How was the data associated with each instance acquired?** The creators used the open-source mathematics software system SageMath to calculate character values.

- **What mechanisms or procedures were used to collect the data (e.g., hardware apparatuses or sensors, manual human curation, software programs, software APIs)?** SageMath was used for the calculations and Python was used to sort, format, split the data.

- **If the dataset is a sample from a larger set, what was the sampling strategy (e.g., deterministic, probabilistic with specific sampling probabilities)?** N/A

- **Who was involved in the data collection process (e.g., students, crowdworkers, contractors) and how were they compensated (e.g., how much were crowdworkers paid)?** Henry Kvinge wrote the code in SageMath to generate this dataset.

- **Was any preprocessing/cleaning/labeling of the data done (e.g., discretization or bucketing, tokenization, part-of-speech tagging, SIFT feature extraction, removal of instances, processing of missing values)?** Partitions were listed as sequences of numbers enclosed by brackets.

- **Has the dataset been used for any tasks already?** No.

- **Is there a repository that links to any or all papers or systems that use the dataset?** Papers that use the dataset will be listed at `https://github.com/pnnl/ML4AlgComb`.

- **What (other) tasks could the dataset be used for?** This dataset could be used for any tasks that aim to understand around the ability to machine learning to perform challenging mathematical tasks.

- **Is there anything about the composition of the dataset or the way it was collected and preprocessed/cleaned/labeled that might impact future uses?** No.

- **Are there tasks for which the dataset should not be used? If so, please provide a description.** No.

- **Will the dataset be distributed to third parties outside of the entity (e.g., company, institution, organization) on behalf of which the dataset was created? If so, please provide a description.** No.

- **How will the dataset will be distributed (e.g., tarball on website, API, GitHub)? Does the dataset have a digital object identifier (DOI)?** All datasets will be compressed to a single `.zip` file and stored on Google Drive. Digital object identifiers will be generated before the paper and datasets are publicly announced.

- **Will the dataset be distributed under a copyright or other intellectual property (IP) license, and/or under applicable terms of use (ToU)?** CC0, `https://creativecommons.org/public-domain/cc0/`

- **Have any third parties imposed IP-based or other restrictions on the data associated with the instances?** No.

## C.9 Weaving patterns

### C.9.1 Context

Weaving patterns can be enriched by replacing the $\{0, 1\}$ entries to the matrix with $\{1, 2, \ldots, n\}$ entries that track the element being swapped. An $O(n^2)$ algorithm for determining if a given 0-1 matrix is a valid weaving pattern exists but gives no additional insight into the structure of weaving patterns and correspondingly the asymptotics of reduced decompositions.

The enumeration of reduced decompositions up to commutation equivalence has been studied by many including Knuth and Stanley. An exact formula is likely out of reach, so asymptotic upper and lower bounds are of great interest. ML models that can detect necessary or sufficient conditions for a matrix to be a valid weaving pattern have the potential to lead to substantial improvements in the upper bound.

**Dataset:** A mixture of enriched weaving patterns and non-weaving pattern matrices with $\{1, 2, \ldots, n\}$-entries.

**Task:** Classify whether a matrix in the dataset is a weaving pattern or not.

### C.9.2 The datasets themselves

Weaving patterns of size $n \times n - 1$ are a special type of matrix containing entries in $\{1, 2, \ldots, n\}$. They correspond to representations of the longest word permutation of $n$ elements (the permutation that sends $1 \mapsto n$, $2 \mapsto n - 1$, etc.). This task involves trying to identify weaving patterns among matrices that look like weaving patterns but are not.

Each $n \times n - 1$ matrix is stored on single line. For instance,

```
(0, 1, 2, 3, 3, 2, 3, 4, 2, 3, 2, 1, 5, 4, 3, 2)
```

where the matrix is represented in row-major format. An integer representing whether the matrix corresponds to a weaving pattern (a '0') or not (a '1') is listed after the matrix.

We provide files for $n = 6, 7, 8$. The files are:

- `weaving_patterns_6_train.txt`
- `weaving_patterns_7_train.txt`
- `weaving_patterns_8_train.txt`
- `weaving_patterns_6_test.txt`
- `weaving_patterns_7_test.txt`
- `weaving_patterns_8_test.txt`

### C.9.3 Weaving patterns datasheet

- **For what purpose was the dataset created?** This dataset was created to study machine learning model's ability to illuminate properties of weaving patterns.

- **Who created the dataset (e.g., which team, research group) and on behalf of which entity (e.g., company, institution, organization)?** The dataset was created by Herman Chau of the University of Washington (true weaving patterns) and Davis Brown of Pacific Northwest National Laboratory (false weaving patterns).

- **What do the instances that comprise the dataset represent (e.g., documents, photos, people, countries)?**
  Instances consist of a $\{1, 2, \ldots, n\}$-valued matrix and a 0 if the matrix is a weaving pattern and a 1 if not.

- **How many instances are there in total (of each type, if appropriate)?** For $n = 6$ there are $2,501$ instances, for $n = 7$ there are $162,000$ instances, for $n = 8$ there are $26,471,025$ instances.

- **Does the dataset contain all possible instances or is it a sample (not necessarily random) of instances from a larger set?** Each dataset contains all weaving patterns. The non-weaving patterns are a sample from all $\{1, 2, \ldots, n\}$-valued matrices obtained by permuting two of the entries in the row of a weaving pattern and checking that the resulting matrix is not a weaving pattern.

- **What data does each instance consist of?** Instances consist of a $\{1, 2, \ldots, n\}$-valued matrix and a 0 if the matrix is a weaving pattern and a 1 if not.

- **Is there a label or target associated with each instance?** The final 0 or 1 in the row is the label.

- **Is any information missing from individual instances?** No.

- **Are relationships between individual instances made explicit (e.g., users' movie ratings, social network links)?** No.

- **How was the data associated with each instance acquired?** Positive examples were generated by a Java script. Negative examples were generated via the positive examples using a Python script.

- **What mechanisms or procedures were used to collect the data (e.g., hardware apparatuses or sensors, manual human curation, software programs, software APIs)?** None.

- **If the dataset is a sample from a larger set, what was the sampling strategy (e.g., deterministic, probabilistic with specific sampling probabilities)?** Each dataset contains all weaving patterns. The non-weaving patterns are a sample from all $\{1, 2, \ldots, n\}$-valued matrices obtained by permuting two of the entries in the row of a weaving pattern and checking that the resulting matrix is not a weaving pattern.

- **Who was involved in the data collection process (e.g., students, crowdworkers, contractors) and how were they compensated (e.g., how much were crowdworkers paid)?** Code was written by Herman Chau of the University of Washington (true weaving patterns) and Davis Brown of Pacific Northwest National Laboratory (false weaving patterns).

- **Was any preprocessing/cleaning/labeling of the data done (e.g., discretization or bucketing, tokenization, part-of-speech tagging, SIFT feature extraction, removal of instances, processing of missing values)?** Matrices were flattened via row-major format. The first and last rows were removed.

- **Has the dataset been used for any tasks already?** No.

- **Is there a repository that links to any or all papers or systems that use the dataset?**

- **What (other) tasks could the dataset be used for?** The dataset could be used for other tasks related to the study of weaving patterns.

- **Is there anything about the composition of the dataset or the way it was collected and preprocessed/cleaned/labeled that might impact future uses?** No.

- **Are there tasks for which the dataset should not be used? If so, please provide a description.** No.

- **Will the dataset be distributed to third parties outside of the entity (e.g., company, institution, organization) on behalf of which the dataset was created? If so, please provide a description.** No.

- **How will the dataset will be distributed (e.g., tarball on website, API, GitHub)? Does the dataset have a digital object identifier (DOI)?** All datasets will be compressed to a single `.zip` file and stored on Google Drive. Digital object identifiers will be generated before the paper and datasets are publicly announced.

- **Will the dataset be distributed under a copyright or other intellectual property (IP) license, and/or under applicable terms of use (ToU)?** CC0, `https://creativecommons.org/public-domain/cc0/`

- **Have any third parties imposed IP-based or other restrictions on the data associated with the instances?** No.

## D   Model Performance

Overall, we found that the models with an MLP architecture tended to perform most consistently across the datasets in this collection (at least given our simple training set up). Transformers performed equivalently to our MLPs on several datasets (Weaving Patterns $n = 6, 7$, Grassmannian Cluster Algebras, and mHeight $n = 10$) and better on one (KL polynomials $n = 9$), but otherwise lagged in performance. This could be due to a number of factors including a hyperparameter optimization set-up that is non-optimal for transformers or the need for larger transformers. We also see the sensitivity bias of transformers [16] as being counter to many problems in combinatorics where the ground truth label can change radically with small changes to the input (at least for many standard representations of combinatorial gadgets). A toy example of this is the parity of a permutation which can easily change with a single change to the representation of a permutation in one-line notation. Developing novel representations of input data that result in a task no longer being sensitive to small changes would be an interesting direction of research.

Unsurprisingly, we found that larger datasets were generally associated with better model performance. This is generally true even in the case where generating a larger dataset required increasing $n$, and thus making the problem more complex (e.g., working with partitions of $n + 1$ rather than partitions of $n$). There were some tasks however that seem hard even when the dataset size is increased. As can be seen in Table 2, performance regressing symmetric group characters is very poor. Indeed, for the largest value of $n$ that we explored, $n = 22$, linear regression performed better than either an MLP or

a transformer. This may relate to the complexity of the task (calculating symmetric group characters is known to belong to $\#P$ [17]). It may also relate to the distribution of symmetric group characters which has a very long tail.

### D.1 Baseline hyperparameters

For our baselines, we train encoder-only transformer models, standard feedforward multi-layer perceptron (MLP) models with ReLU non-linearities, and logistic regression on the classification tasks and the same architectures of transformers and MLPs along with linear regression for the regression tasks.

- To optimize MLP models we performed a simple grid search across
  - learning rates (0.001, 0.0005, and 0.0001),
  - depths (1, 2, 3, and 4),
  - and constant hidden dimension (32, 64, 128, and 256).
- To optimize the transformer models we performed a similar grid search but with hyperparameters
  - learning rates (0.001, 0.0005, and 0.0001),
  - model dimensionality (20, 40, and 80),
  - depths (2 and 4),
  - and the number of heads (4, 6, and 8).

All optimization was performed in Pytorch with the Adam optimizer on one Nvidia $A100$. For each hyperparameter setting we trained 3 models and averaged their performance over the resulting values. We use these to generate the $95\%$ confidence intervals found on Table 1.

Linear and logistic regression was performed with sklearn using standard hyperparameters.

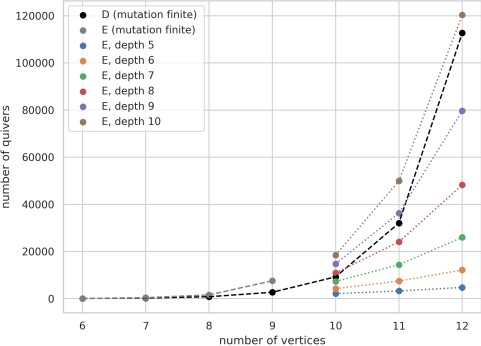

Figure 8: The growth in examples in the *Mutation Equivalence of Quivers Dataset* as depth and number of vertices is increased

| Dataset | # of points | # of classes | Length of input seq | # of tokens | Task |
| --- | --- | --- | --- | --- | --- |
| Lattice paths | | | | | |
| $n = 10$ | 14,296 | 2 | 38 | 2 | Classification |
| $n = 11$ | 49,807 | 2 | 42 | 2 | Classification |
| $n = 12$ | 175,267 | 2 | 46 | 2 | Classification |
| $n = 13$ | 621,738 | 2 | 50 | 2 | Classification |
| Weaving patterns | | | | | |
| $n = 6$ | 2,501 | 2 | 16 | 6 | Classification |
| $n = 7$ | 162,000 | 2 | 25 | 7 | Classification |
| $n = 8$ | 26,471,025 | 2 | 36 | 8 | Classification |
| Cluster algebra quivers | | | | | |
| $n = 11$ | 204,739 | 7 | 120 | 6 | Classification |
| $S_n$ characters | | | | | |
| $n = 18$ | 140,846 | NA | 36 | 19 | Regression |
| $n = 20$ | 373,480 | NA | 40 | 21 | Regression |
| $n = 22$ | 953,835 | NA | 44 | 23 | Regression |
| KL Polynomials | | | | | |
| $n = 8$ | 84,624 | 10 | 16 | 8 | Classification |
| $n = 9$ | 204,739 | 10 | 18 | 9 | Classification |
| mHeight function | | | | | |
| $n = 10$ | 467,627 | 5 | 45 | 2 | Classification |
| $n = 11$ | 3,283,963 | 5 | 55 | 2 | Classification |
| $n = 12$ | 23,079,881 | 5 | 66 | 2 | Classification |
| Schubert polynomials | | | | | |
| $n = 4$ | 2,105 | 2 | 15 | 7 | Classification |
| $n = 5$ | 107,025 | 3 | 19 | 9 | Classification |
| $n = 6$ | 10,508,205 | 5 | 23 | 11 | Classification |
| RSK | | | | | |
| $n = 8$ | 40,320 | NA | 54 | 11 | Permutation prediction |
| $n = 9$ | 362,880 | NA | 60 | 12 | Permutation prediction |
| $n = 10$ | 3,628,800 | NA | 66 | 13 | Permutation prediction |
| Grassmannian cluster algebras | | | | | |
| $n = 6$ | 185,822 | 2 | 12 | 12 | Classification |