# OpenReview forum: "Machine Learning meets Algebraic Combinatorics: A Suite of Datasets to Accelerate AI for Mathematics Research"
_NeurIPS.cc/2024/Workshop/MATH-AI — MATH-AI 24_

### Official Review · Reviewer_9r5x · 2024-10-05
**A Practical Dataset Bridging AI and Algebraic Combinatorics**

**Rating:** 10
**Confidence:** 4

**Review:**

This paper introduces a novel collection of datasets aimed at facilitating machine learning (ML) applications in algebraic combinatorics. The comprehensive suite of nine datasets, each tackling classic or open problems in algebraic combinatorics is highly interconnected with representation theory and algebraic geometry. The primary motivation is to bridge the gap between machine learning and advanced mathematics by providing structured datasets designed to assist in scientific discovery. The dataset includes tasks ranging from predicting permutation properties, such as the mHeight function, to identifying patterns in combinatorial structures like quivers and lattice paths. The datasets cover both well-studied mathematical results (e.g., symmetric group characters and the Robinson-Schensted-Knuth correspondence) and open problems (e.g., weaving patterns and Kazhdan-Lusztig polynomials). The paper also discusses baseline models, such as logistic regression, multilayer perceptrons (MLPs), and transformers, demonstrating varying performance across tasks.

The authors carefully curated datasets based on both open and classical mathematical problems, ensuring a balance between mathematical rigor and accessibility for the ML community. Besides, the datasets are described in great detail, from their mathematical context to the specific ML tasks associated with them. The inclusion of background sections and summaries for each dataset makes the paper approachable for both mathematicians and AI researchers with limited exposure to algebraic combinatorics. Additionally, by targeting both the ML and mathematics communities, this research promotes interdisciplinary collaboration. It offers interpretability researchers an opportunity to apply ML methods to inherently difficult problems, such as mutation equivalence of quivers or partial orders on lattice paths, providing a fertile ground for future research.

---

### Official Review · Reviewer_VmKL · 2024-10-07
**Interesting New Dataset for the Mathematics**

**Rating:** 7
**Confidence:** 3

**Review:**

This paper introduces a dataset comprised of multiple problems in Algebraic Geometry, some of which are open problems in the mathematics community. The purpose of the dataset is to be able to advance research in mathematics, aid mathematicians in their research, and train ML models on mathematical reasoning tasks. The dataset includes 9 different mathematics tasks and includes both classification and regression tasks. The datasets are assessed with linear regression, an MLP, and a transformer.

This is certainly a benefit to the mathematics and ML communities. The authors also took care to discuss limitations of the data and challenges in creating a good variety and class-balanced data. This is not a critique but more of something to note: It would have been interesting to see if ML could have discovered any interesting properties on any of the open problems. I realise this is a hard task and could be its own paper, but it would certainly show the benefit of this dataset.

Minor comments:

-	Line 17: space between constructing and “good”

-	Line 23: cite some examples on how ML has aided mathematicians in their research

-	Figure 1: Would be nice to see what the input is like that directly goes into the ML model. For e.g. its not entirely obvious for cluster algebra quivers if the input is a graph object for graph neural networks or simply a list of integers?

-	Some examples use a small amount of data for training (e.g. quivers dataset for 11x11 matrices). Perhaps its because I don’t have specific knowledge for this problem but is it not possible to create more data than roughly 20k example? All models would certainly benefit from more data.

---

### Official Review · Reviewer_6zsG · 2024-10-07
**A Promising Step Towards AI for Mathematics: A Review of the Algebraic Combinatorics Dataset Repository**

**Rating:** 7
**Confidence:** 4

**Review:**

Summary: The paper "Machine Learning meets Algebraic Combinatorics: A Suite of Datasets to Accelerate AI for Mathematics Research" presents a collection of datasets designed to facilitate the development of machine learning methods for advancing research-level mathematics. The authors introduce the Algebraic Combinatorics Dataset Repository (ACD Repo), which includes nine datasets representing either classic or open problems in algebraic combinatorics. The tasks associated with each dataset are described, and the challenges involved in generating useful mathematics datasets are discussed. The authors also provide baseline model performance results using logistic regression, MLPs, and encoder-only transformers. The paper concludes by highlighting the limitations of the datasets and emphasizing the potential of machine learning tools for mathematics research.

Pros:
Comprehensive dataset collection: The paper presents a diverse and well-structured collection of datasets, covering various problems in algebraic combinatorics.
Clear task descriptions: Each dataset is accompanied by a clear and concise task description, making it easy for researchers to understand the problem and start working on it.
Baseline model performance: The authors provide baseline model performance results, which can serve as a starting point for future research and help evaluate the effectiveness of new approaches.
Discussion of challenges: The paper highlights the challenges involved in generating useful mathematics datasets, which can help raise awareness and inspire further research in this area.
Cons:
Limited dataset size: Some of the datasets may be relatively small, which could limit their usefulness for training and evaluating machine learning models.
Lack of diversity in tasks: While the datasets cover various problems in algebraic combinatorics, they may not represent the full range of problems and tasks that researchers might be interested in exploring.
No explicit evaluation metrics: The paper does not provide explicit evaluation metrics for each task, which could make it difficult for researchers to compare and evaluate different approaches.

---

### Decision · Program_Chairs · 2024-10-08

**Decision:**

Accept

**Comment:**

The reviewers are generally positive about the work, and it makes for a very interesting contribution in our workshop.